# Safety and Efficacy of New Oximes to Reverse Low Dose Diethyl-Paraoxon-Induced Ventilatory Effects in Rats

**DOI:** 10.3390/molecules25133056

**Published:** 2020-07-03

**Authors:** Maya Kayouka, Pascal Houzé, Marc Lejay, Frédéric J. Baud, Kamil Kuca

**Affiliations:** 1Bioactive Molecules Research Laboratory, Faculty of Sciences, Section II, Lebanese University, Beirut, Lebanon; maya.kayouka@ul.edu.lb; 2Laboratoire de Toxicologie Biologique, GH Lariboisière-St Louis-F. Widal, Assistance Publique-Hôpitaux de Paris (AP-HP), 75010 Paris, France; 3Unité de Technologies Chimiques et Biologiques pour la Santé (UTCBS), CNRS UMR8258—U1022, Faculté de Pharmacie Paris Descartes, Université de Paris, 75006 Paris, France; 4Département d’Anesthésie—Réanimation-SAMU de Paris, Hôpital Universitaire Necker-Enfants malades, 75015 Paris, France; marc.lejay@aphp.fr (M.L.); baud.frederic@wanadoo.fr (F.J.B.); 5EA 7323, Pharmacologie et évaluations thérapeutiques chez l’enfant et la femme enceinte, Université de Paris, F-75006 Paris, France; 6Department of Chemistry, Faculty of Science, University of Hradec Kralove, 50003 Hradec Kralove, Czech Republic; kamil.kuca@uhk.cz

**Keywords:** pralidoxime, KO-27, BI-6, oximes, rats, plethysmography, diethyl-paraoxon, ventilatory effects

## Abstract

Background: Oximes are used in addition to atropine to treat organophosphate poisoning. However, the efficiency of oximes is still a matter of debate. In vitro experiments suggested than new oximes are more potent than the commercial oximes. However, the antidotal activity of new oximes has not been assessed in vivo. Methods: The aim of this work was to assess the safety and efficiency of new oximes compared to pralidoxime in a rat model of diethyl paraoxon-induced non-lethal respiratory toxicity. Results: Safety study of oximes showed no adverse effects on ventilation in rats. KO-33, KO-48, KO-74 oximes did not exhibit significant antidotal effect in vivo. In contrast, KO-27 and BI-6 showed evidence of antidotal activity by normalization of respiratory frequency and respiratory times. KO-27 became inefficient only during the last 30 min of the study. In contrast, pralidoxime demonstrated to be inefficient at 30 min post injection. Inversely, the antidotal activity of BI-6 occurred lately, within the last 90 min post injection. Conclusion: This study showed respiratory safety of new oximes. Regarding, the efficiency, KO-27 revealed to be a rapid acting antidote toward diethylparaoxon-induced respiratory toxicity, meanwhile BI-6 was a late-acting antidote. Simultaneous administration of these two oximes might result in a complete and prolonged antidotal efficiency.

## 1. Introduction

Organophosphates (OPs) are used daily around the world as pesticides. However, they remain a major health concern because of the large annual number of acute poisonings. Indeed, according to the data of the World Health Organization, there are more than 3 million of organophosphate intoxications annually and more than 220,000 deaths [1,2].

Respiratory failure is considered the primary cause of death in acute organophosphate poisoning [3,4,5]. Still, the exact mechanism remains unclear because of a combination of peripheral effects like constriction and increased secretion in the airways, paralysis of the respiratory musculature, and a direct depressant effect on the respiratory centers in the brainstem [6]. The standard therapeutic scheme of insecticide poisoning includes supportive treatment, decontamination, and administration of antidotes including atropine and oximes [7]. However, the respective antidotal role of atropine and pralidoxime still remains a pending question. In human poisoning, large cumulative doses of atropine are advised as first line treatment [8,9,10]. The efficiency of atropine has been clearly demonstrated [11].

There has been a recent regain of interest in assessing the efficiency of new oximes, compared to older products, in reversing OPs toxicity [12,13]. Oximes are nucleophilic agents allowing the reactivation of acetylcholinesterase by removal of the phosphoryl group [13]. Since the early fifties, pralidoxime (PRX) (Figure 1) and obidoxime have been used worldwide in the treatment of OPs poisonings [10]. However, to our knowledge, randomized clinical trials were performed only using PRX salts [5,7,14]. Owing to the conflicting results of these trials, the efficiency of PRX is still questioned in the treatment of acute OPs poisoning. Indeed, multiple meta-analysis as well as several Cochrane reviews failed to evidence PRX efficacy [15,16]. However, these meta-analyses always dealt with the same set of studies.

We developed a rat model to assess ventilatory effects induced by a non-lethal model of paraoxon toxicity using whole body plethysmography in awake animals [17,18]. Noteworthy, the efficiency of PRX was tested alone, without administrating any dose of atropine [19,20]. This model was shown to be highly reproducible using different species, including rats and mice, and different inhibitor of cholinesterase like OPs and carbamate [21,22,23,24].

Using PRX as a reference, the aim of this study was to compare the safety and the efficiency of a selection of new promising oximes [25] namely KO-27, KO-33, KO-48, KO-74, and BI-6 (Figure 1). Thus, detailed comparison with PRX was performed by the intermediate in our rat model of non-lethal model of paraoxon toxicity.

## 2. Results

### 2.1. Study 1: Safety of New Oximes on Ventilation Parameters at Rest in Awaken and Unrestrained Rats

Baseline values: There were no significant differences regarding to the temperature, the frequency (f), total time (T_TOT_), expiratory time (T_E_), inspiratory time (T_I_), tidal volume (V_T_), and minute volume (V_E_) when comparing the control and oxime groups.

Clinical findings: Compared to the control group, the rats treated using the different oximes exhibited neither clinical signs nor abnormal behavior.

During the study, the core temperature in the control group remained normal and stable. The administration of oximes did not induce statistically significant effects on the core temperature, as reported in Table 1.

At the end of the study, the mean temperature in the control, PRX, KO-27, KO-33, KO-48, KO-74, and BI-6 groups were 37.6 ± 0.1 °C, 38.3 **±** 0.2 °C, 37.9 **±** 0.3 °C, 38.1 **±** 0.3 °C, 38.4 **±** 0.2, 37.8 **±** 0.1 °C, and 37.9 ± 0.3 °C, respectively.

Plethysmography: As emphasized in Table 2, no significant variations were observed for the different respiratory parameters after intra-muscular injection of the different oximes, compared to those in the control group.

### 2.2. Study 2: Efficiency of Novel Oximes on Ventilation Parameters at Rest in Awaken and Unrestrained Diethyl-Paraoxon-Poisoned Rats

Baseline values: There were no significant differences of the temperature, f, T_TOT_, T_E_, T_I_, V_T_, and V_E_ when comparing the control and oximes groups.

Clinical findings: All the rats poisoned with diethyl-paraoxon (DEPO) exhibited urination, defecation, hypersalivation, and fasciculation. The mean core temperatures in the control group remained unchanged. After DEPO administration, the mean core temperatures decreased in the DEPO group and were significantly lower than the values observed for the control group from 30 min until the completion of the study (*p* < 0.001) (Figure 2).

As reported in Table 3, the administration of PRX or new oximes did not have any effect on hypothermia induced by subcutaneous injection of DEPO. All oxime-treated groups exhibited significantly different core temperatures compared to the control group.

Compared to PRX, only BI-6 injection enabled a significant and nearly complete correction of core temperature at the end of the study period. In the DEPO+BI-6 group, the mean temperature decreased and remained low until 120 min; later on the temperature increased significantly (*p* < 0.01) and demonstrated to be significantly higher at the end of the experiment (210 min) in comparison with the DEPO group (*p* < 0.001) (Figure 2). However, the results did not show any significant differences between the administration of PRX and BI-6 on the temperature values at 120- and 150-min post-injection of oxime.

Plethysmography: The administration of DEPO induced the rapid onset of significant alterations of the ventilatory parameters (Figure 3). In comparison to the control group, the *T*_TOT_ and *T*_E_ increased and plateaued 30 min after DEPO administration until the completion of the study (*p* < 0.001). The *V*_T_ also increased significantly and plateaued 30 min after DEPO administration (*p* < 0.05, *p* < 0.01 or *p* < 0.001) but decreased at time 180 and 210 min. Conversely *f* decreased in the DEPO group and was significantly lower than the control group (*p* < 0.001). DEPO did not have any significant effect on *T*_I_ and *V*_E_.

In comparison with DEPO group, the DEPO+PRX group showed a significant decrease in *T*_TOT_ between 15 and 30 min consequently to PRX administration (*p* < 0.05 or *p* < 0.01) and in *T*_E_ between 5 and 45 min (*p* < 0.01 or *p* < 0.001). A significant increase in *f* could be observed between 5 and 45 min after PRX administration (*p* < 0.01 or *p* < 0.001). Regarding *V*_T_, a significant decrease was effectively recorded from 15 to 90 min after PRX administration (Figure 3). Furthermore, the values of the different parameters could not return to the normal range. Finally, PRX did not have any significant effect on *T*_I_ and *V*_E_.

In comparison with the DEPO group, the DEPO+KO-27 group showed a significant decrease in *T*_TOT_ from 150 to 180 min following KO-27 administration (*p* < 0.05 or *p* < 0.001). *T*_E_ also showed a significant decrease from 15 to 180 min after KO-27 administration (*p* < 0.001), similarly to *f* which increased as well (*p* < 0.05 or *p* < 0.001). During this period of time, *f* returned within the normal range. KO-27 did not have any significant influence on *V*_T_*, T*_I_, and *V*_E_.

Compared to DEPO group, the DEPO+BI-6 group showed a significant decrease in *T*_TOT_ and *T*_E_ between 60 and 180 min after BI-6 administration (*p* < 0.01 or *p* < 0.001). In parallel, data obtained showed a significant increase in *f* between 120 and 180 min after BI-6 administration (*p* < 0.001). During this period, *T*_TOT_, *T*_E_, and *f* returned to values within and above the normal range. Regarding *V*_T_, a significant decrease could be observed from 60 to 180 min after BI-6 administration (*p* < 0.001). During this period, *V*_T_ returned in the normal range (Figure 3). Finally, the administration of BI-6 did not result in any significant effect on *T*_I_ and *V*_E_.

As reported in Table 4, the new oximes assessed (KO-33, KO-48, and KO-74) did not induce any statistically significant effects on respiratory parameters altered by the subcutaneous injection of DEPO.

## 3. Discussion

The past record regarding involvement of organophosphorus compounds in poisoning clearly indicates the different conditions the health system may face. Acute household OPs poisonings require to address decontamination, supportive treatment, and antidotal treatment. Meanwhile, the prevention of household poisonings should also be considered as a major issue. In contrast, counterterrorism requires to protect soldiers that might be exposed to nerve agents. Therefore, carbamates derivatives exhibiting strong anticholineresterasic effects have been used extensively to treat a number of diseases. They include for instance myasthenia gravis which requires long-term treatment as well as short-term treatment to reverse muscarinic effects induced by depolarizating myorelaxants at the end of anaesthesia. However, anticholinesterasic carbamates are antagonist antidotes with intrinsic effects. While attempting to prevent uncertain exposure to nerve agents, the occurrence of the intrinsic effects resulted in partial incapacitation of soldiers, pre-treated with carbamate without subsequent exposure to nerve agents [26].

The basic treatment of overt acute organophosphorus poisoning is based on administration of high flow of normobaric oxygen immediately followed by repeated administration of important doses of atropine. Especially, atropine doses are usually superior than 2 mg in adults administered by intravenous injection every 3 minutes until reversal of respiratory effects [27]. By themselves, marked muscarinic effects may result in potentially life-threatening respiratory distress and even respiratory failure. Muscarinic respiratory toxicity results from a combination of profuse intractable hypersecretion of saliva and bronchorrhea, combined with constriction of the white muscles in the bronchial tree. The combination of airways obstruction by profuse secretion and narrowing of bronchial lumen may [28] therefore result in acute respiratory failure. The administration of atropine at appropriate doses is always efficient. Because of its pharmacokinetics compared to the duration of muscarinic effects induced by nerve agents on the bronchial tree, one frequent issue is the occurrence of atropine overdose at a time when muscarinic effects completely disappeared [29].

Oximes are thus acting as reactivators of cholinesterases, inhibited by nerve agents, which represents a far more physio-pathological treatment than atropine. Indeed, reactivation of cholinesterase using atropine is expected to reverse muscarinic as well as nicotinic effects, including fasciculation of peripheral muscles and diaphragmatic paralysis. Still to this point, oximes are considered to be a second line treatment in addition to atropine. Regarding the oximes, the pharmacodynamic concern includes different in vitro efficiency to counteract nerve agent toxicity. The pharmacokinetic concern identified using PRX in rats, without and with OP poisoning, is a rapid elimination from the body by the kidneys because of active excretion in urine by organic cation transporter [19,20]. Although PRX was shown to rapidly and completely reverse paraoxon-induced respiratory toxicity in rats, its antidotal effect lasted only 30 min meanwhile paraoxon toxicity lasted more than 6 h [30]. A pharmacokinetic study in a guinea pig model reported half-life clearance of obidoxime in the range of 30 min meanwhile the clearance of PRX was in the range of 20 min [31]. The route of administration is a major bias when determining the true elimination half-life. In a previous study, we showed the elimination half-life of PRX administered by intravenous route in the rat was 29.4 ± 1.1 min and 54 ± 9.3 min after intramuscular administration [30]. Bohnert et al. [31] reported elimination half-life of PRX of 18.9 ± 9.5 min, after intramuscular injection using auto-injector device. Therefore, the values in rats seem to be different from experiments using the guinea pigs. Unfortunately, the authors did not compare their results to ours. A shorter half-life after intramuscular administration in comparison with bolus intravenous administration questions about the device used for administration. Thereby, it would have been interesting if the authors studied the elimination of a bolus intravenous dose of PRX which still remains the gold standard for measuring true elimination half-life. Nonetheless, both available oximes exhibit very rapid elimination requiring either repeated injections or continuous infusion that showed to be efficient in humans severely poisoned with OPs series using obidoxime [32] as well as in a controlled study using PRX [5].

The other randomized clinical trial which did not report clinical efficacy of PRX in humans poisoned with OP insecticides, dealt with non-severe patients that did not allow assessing the efficacy of PRX. Indeed, the percentages of patients intubated at baseline in the control and PRX groups reported in Eddleston et al. [7] and Pawar et al. [5] studies were 14 versus 69% and 19.8 versus 63%, respectively. In the case series, several patients exhibited the elimination half-life reported superior to 10 h which appears to be very long compared to the values reported in other patients [32]. No explanation was provided to attribute this discrepancy. Obidoxime was reported to be eliminated by renal route [32] similarly to PRX. Albeit in this case series, urine concentrations of obidoxime were not reported [32]. The extended elimination half-life measured in this patient could be explain by an impairment of the renal function. Also, the administration of non-specific substrate of organic cation transporter (OCT), such as thiamine, could have markedly reinforced the antidotal activity of obidoxime. Indeed, in non-physiological or physiological models, we showed that both potassium dichromate-induced alteration of renal function as well as substrates of OCT 1,2, and 3 increased elimination half-life of PRX that closely paralleled the increase in antidotal activity [2,19,20]. Furthermore, we showed that substrates of OCT1 and 2, on contrary to OCT3, altered PRX elimination and increased its antidotal activity [19].

The present in vivo study showed that none of the six tested oximes exhibited any intrinsic effects on the respiratory system at rest. Furthermore, the KO-48 oxime was previously shown to be devoid of any significant adverse effects, including stimulation of oxidative stress markers, cytotoxicity, and genotoxicity in rats exposed to the oxime at 25% of its LD_50_ [33].

Unfortunately, our rat model did not evidence any antidotal activity of KO-48 toward PO-induced respiratory toxicity. This finding supports the hypothesis that the rat model clearly emphasized that cholinesterase reactivation in vitro is not predictive of an antidotal efficacy in vivo. As a matter of fact, among the six oximes tested, only three evidenced significant antidotal efficacy in vivo, including PRX, KO-27, and BI-6 in reversing overt DEPO-induced respiratory toxicity. One limitation of our study results from the absence of assessment regarding the dose-effect relationship for each oxime. Therefore, we cannot exclude any significant antidotal efficacy of the other apparently inefficient oximes at doses greater than those tested in our study. Indeed, Petroianu et al., reported that KO-48 and KO-27 were the most promising oximes as they could be associated with an improvement in survival of rats poisoned with PO [34]. However, in Lorke’s study the antidote was administered 1 min after PO. Therefore, this design enables for the protective effect of the antidote to be assessed rather than the ability to reverse overt toxicity based on the relative risk of death after administration of different OPs at three doses [35]. Thus, in our study dealing with respiratory toxicity, the oximes were administered at the onset of maximal respiratory toxicity, 30 min after DEPO subcutaneous injection, similarly to our previous studies dealing with the antidotal activity of PRX [2,19,20].

The present study confirmed the early and complete efficiency of PRX in reversing PO-induced toxicity albeit this antidotal effect was of relatively short duration [30]. The rat model unveiled similarities and differences in the antidotal efficacy of KO-27 and BI-6 in comparison to PRX. KO-27 acts as rapidly as PRX in reversal PO-induced toxicity, within minutes after injection. However, in contrast with the short reversal effect in DEPO toxicity induced by PRX which lasted 20 min post-injection, the effect of a single dose of KO-27 demonstrated to last 180 min post-injection. In contrast, BI-6 appeared to be devoid of any early protective effects toward DEPO toxicity from 30 to 90 min post-injection meanwhile its antidotal activity became statistically significant after 90 min.

The potential consequences of the results of our study are manifold.

Owing to the rapid onset and long-lasting effect of KO-27, the results suggest that one single dose of KO-27 may relevantly replace the continuous or repeated injections of PRX in a context where healthcare resources are limited. However, this proposal is hampered by the lack of data supporting the safety of KO-27 in human.Figure 3—Panel B suggests that a combination of PRX and BI-6 may reverse DEPO-induced increase of expiratory time during the study period of 210 min. Indeed, PRX led to a rapid reversal of expiratory time increase consequently to PRX injection without any noticeable effect of BI-6. Thereafter, PRX lasted at the time the antidotal effect of BI-6 occurred at a time when PRX antidotal effect disappeared with long-lasting antidotal of BI-6 and later increase until the end of the experiment. As expected, all altered respiratory parameters improved in the same manner. These data suggest that a combination of PRX and BI-6 should also be considered in humans. However, this proposal is hampered by the absence of knowledge regarding to the pharmacokinetic drug–drug interaction in addition to the elimination of PRX and BI-6 administered at the same time. Also the lack of data supporting the safety of BI-6 in humans requires to be addressed.

These findings suggest that the combination of a single dose of both oximes administered at the same time, may induce long-lasting reversal of DEPO-induced toxicity. Indeed, as we previously reported when dealing with both toxicity and the reversal effect by antidote, it is mandatory to consider not only the magnitude of the effect but also, its time-course [36]. We should conclude that KO-27 and BI-6 evidence the same potency to completely reverse DEPO-induced toxicity with different toxicodynetics. However, KO-27 revealed to be acting early during the poisoning meanwhile BI-6 is acting later and protracting the whole antidotal efficacy of both oximes. One limitation of our study results from the fact that we did not test the co-administration of KO-27 and BI-6 to assess additivity of antidotal effects in our rat model.

Furthermore, experimental studies allow testing the drugs in conditions that would be unlikely when facing human poisonings. Indeed, our rat model showed that one single dose of oxime without any associated treatment with atropine consistently resulted in the reversal of DEPO-induced respiratory toxicity with short duration of action for PRX while being long-lasting with KO-27 and BI-6. In human treatment, atropine would be mandatory. These experimental studies support the hypothesis that oxime, including not only PRX but also KO-27 and BI-6 may result in a sparing effect on the cumulative doses of atropine required to treat poisoning.

Our results showing complete reversal by an oxime or a combination of oximes suggest two major improvements in the treatment of OP poisoned patients, especially involving nerve agents that may cause a huge number of victims: less need of human resources and sparing effect on atropine dose by either KO-27 alone or the combination of PRX and BI-6.

The present results and the safety study support the hypothesis that KO-27, a safe and early-occurring and long-lasting antidote might be considered as a valuable agent for prophylactic treatment toward an ongoing risk of exposure to nerve agents. This hypothesis is further supported by Lorke’s study who showed that the best outcome of DEPO-induced lethal poisonings in rats was achieved when KO-27 was administered prophylactically [37].

The present study suffers from several limitations. The antidotal activity of the different oximes was tested toward only a single OP, paraoxon. Therefore, caution is needed to extend the present results to other OPs, including nerve agents. This study was performed in rats. The toxicity of organophosphate compounds is variable in species depending on different factors. Among these factors, the concentration of different esterases in the serum was an important factor. In rats, the higher carboxyesterase concentration than in humans allowed better protection against organophosphate poisonings. Therefore, caution is needed to extent the present results to other species, including human. The critical point of this study is the calculation of equipotent doses of PRX analogues. Owing to previous studies, the 50 mg/Kg dose of PRX cation was the dose resulting in the maximal antidotal effect while the plasma concentration remained above the classic threshold of 4 mg/L [18,19,20]. This 50 mg/Kg dose was used in this study for the administration of other novel oximes. So, as administered dose is not expressed in mole/Kg, these compounds were not administered at equimolar doses of PRX cation. KO-27 and BI-6 are mono-oximes similarly to PRX and so, have the same number of oxime radicals per molecule. However, the molecular weight of these oximes is about two time that of PRX. As a consequence, they were administered at about the half-molar dose of PRX. Injected at lower molar dose than PRX, these compounds showed to be able to reverse DEPO-induced respiratory toxicity.

In conclusion, the plethysmography method allowed comparing the efficacy of different oximes on toxic non-lethal diethyl-paraoxon-induced ventilatory effects in rats. Using this model, we showed that oximes, including KO-27 and BI-6, developed by Kuca’s team were more potent than PRX to counteract the respiratory effects at rest of DEPO. KO-27 is particularly interesting because of its long-time effect, at least 130 min after intra-muscular administration. At the opposite, BI-6 showed a long-lasting effect and became efficient when the antidotal effect of PRX disappeared. The present results suggest that a combination of PRX and BI-6 therapy should be considered to treat human organophosphate poisoning. The association may even be expected to result in a sparing effect of atropine in reversal over OP toxicity in addition to protect individuals expected to be exposed to nerve agents.

## 4. Material and Methods

All animal procedures used in this study were in strict accordance with the European Community Council Directive of 24 November 1986 (86-609/EEC) (protection of animals used for experimental and other scientific purposes) and Decree of 20 October, 1987 (87-848/EEC) and approved by local ethic Committee under the number is P2.FB.164.10.

### 4.1. Animals and Housing Conditions

Male Sprague-Dawley rats (250–300 g) were purchased from Janvier (Le Genest-St-Isle, France). All the animals were housed in a room with controlled environment (22 ± 3 °C, 55 ± 10% relative humidity), and maintained under a 12-h light/dark cycle. Animals had free access to food and tap water.

### 4.2. Chemicals and Drugs

Diethyl-paraoxon (diethyl p-nitrophenyl phosphate, 90% purity, CAS number: 311-45-5) was obtained from Sigma-Aldrich (St Quentin Fallavier, France). Diethyl-paraoxon was diluted in dimethylsulfoxide (DMSO) so as to obtain a mother solution of 3.5 mg/ml. A daughter solution of paraoxon was prepared in saline (140 µg/mL), so as to inject doses equal to 50% of the LD_50_ as previously described in [17].

Pralidoxime (CAS number: 6735-59-7) methylsulfate (Contrathion^®^) was kindly provided by S.E.R.B laboratory (Paris, France). Pralidoxime was diluted in isotonic saline solution to obtain a 1.0 mg/mL mother solution, stored at −20 °C and stable for a maximum of 6 months. All doses were expressed as PRX cation and a correction factor of 1.7 between PRX and Contrathion^®^ can be used [2,19,20].

New oximes (KO-27, KO-33, KO-48, KO-74, and BI-6) were synthesized in Czech Republic and were a gift of Pr. Kuca (Department of Chemistry, Faculty of Science, University of Hradec Kralove, Hradec Kralove 50003, Czech Republic). Oximes were diluted in isotonic saline solution to obtain a 1.0 mg/mL mother solution, stored at −20 °C, and stable for a maximum of 6 months. PRX and new oximes, were 50 mg/Kg of oxime cation administered intra muscularly. The solutions were stored at +4°C for one-month maximum.

### 4.3. Safety Precaution

All solutions of diethyl-paraoxon were prepared under fume hood using nitrile gloves, overall and goggles. Molar sodium carbonate solution was set under fume hood to neutralize immediately diethyl-paraoxon in the case of accidental spillage. Daily, after each experimental study, the same solution was used to decontaminate and clean up all materials and areas.

### 4.4. Clinical Examination

The animals were clinically observed while plethysmography measurements were performed. The signs were noted and quantified according to De Candole et al. [38]. Clinical examination was semi-quantitative and always realized by the same experimenter.

### 4.5. Whole Body Plethysmography

Ventilatory parameters were recorded in a whole-body plethysmography by the barometric method described and validated in the rat by Bartlett and Tenney [39] with minor modifications, as previously reported [17,18].

The animals were placed in a rectangular Plexiglas chamber with a volume of 3 L connected to a reference chamber of the same size by a high-resistance leak to minimize the effect of pressure changes in the experimental room. The animal chamber was flushed continuously with moisturized air at a rate of 5 L/min. During recording periods, the inlet and outlet tubes were temporarily clamped and the pressure changes associated with each breath were recorded with a differential pressure transducer (Validyne MP, 45 ± 3 cm H_2_O, Northridge, CA, USA), connected to the animal and reference chambers. During each measurement, calibration was performed by one injection of 1 mL of air into the chamber and the ambient temperature was noted. The spirogram was recorded and stored on a computer with acquisition data card (PCI-DAS1000, Dipsi, Chatillon, France) using respiratory acquisition software (Elphy Software, CNRS-UNIC, Gif-sur-Yvette, France) for off-line analysis. This technique was validated according criteria previously reported [17,18].

The following parameters were measured: ambient temperature in the chamber, temperature of the animal, tidal volume (*V*_T,_ expressed in µL), inspiratory time (*T*_I_ expressed in s), expiratory time (*T*_E_ expressed in s) total respiratory time (*T*_TOT_ = *T*_I_ + *T*_E,_ expressed in s), the respiratory frequency (*f* expressed in bat/min), and the minute ventilation (*V*_E_ = *V*_T_ × *f* expressed in µL).

### 4.6. Telemetry Applied to Plethysmography

Central temperature of the rats was measured using infrared telemetry. Rats were anaesthetized with ketamine/xylazine at doses of 70/10 mg/kg respectively, administered intraperitoneally. The infrared telemetry probe (TA10TA-F20, Data Sciences International, Harvard, MA, USA) was inserted aseptically into the peritoneal cavity. The abdomen was closed with sutures and wound clips. The animals were allowed to recover for at least 7 days prior to experimentation [2,19,20].

### 4.7. Study Designs

#### 4.7.1. Study 1: Safety of New Oximes on Ventilation Parameters at Rest in Awaken and Unrestrained Rats

Animals were randomly divided into seven groups (5 animals in each group). The control group received 0.5 mL sodium isotonic solution as solvent of diethyl-paraoxon followed by 0.5 mL sodium isotonic solution as solvent of oximes. The oxime groups received the solvent of diethyl-paraoxon (0.5 mL) followed by PRX, KO-27, KO-33, KO-48, KO-74, and BI-6 solutions using a similar dose to 50 mg/Kg, as stated in the method section. All oximes were administered intramuscularly in the upper of the tight muscle, 30 min after the solvent of diethyl-paraoxon. The first plethysmography measurement was performed after a period of accommodation of 30 min. Then, the rat was gently removed from the chamber for the subcutaneous injection of solvent of diethyl-paraoxon and replaced in the chamber for another session of respiratory recording. Ventilation was recorded 15 and 30 min after injection. Then, the animal was gently removed from the chamber for the intramuscular injection either solvent of oximes for control group, or oxime solutions for oxime groups. The rat was then replaced in the chamber and measurements were recorded at 35, 45, 50, 60, 75, 90, 120, 180, and 210 min

At the end of each experiment, rats were sacrificed with an overdose of pentobarbital.

#### 4.7.2. Study 2: Efficiency of New Oximes on Ventilation Parameters at Rest in Awaken and Unrestrained Diethyl-Paraoxon-Poisoned Rats

Animals were randomly divided into eight groups (8 animals in each group). The control group received 0.5 mL sodium isotonic solution as solvent of diethyl-paraoxon followed by 0.5 mL sodium isotonic solution as solvent of oximes. The poisoned group received diethyl-paraoxon followed by the solvent of oximes. Diethyl-paraoxon was administered subcutaneously at a 0.215 mg/kg dose corresponding to 50% of the LD_50_ determined in our laboratory [17]. A previous study showed that the maximum effect of diethyl-paraoxon on ventilation at rest were observed 30 min post-injection [17]. Therefore, oximes were administered intramuscularly 30 min after diethyl-paraoxon. The rat was then replaced in the chamber and measurements were recorded at 35, 45, 60, 75, 90, 120, 180, and 210 min. At the end of each experiment, rats were sacrificed with an overdose of pentobarbital.

### 4.8. Statistical Analysis

Results are expressed as mean +/− S.E.M (standard error of the mean). Graphs and statistical analysis were performed using Prism version 5.0, GraphPad Software (San Diego, CA, USA). All tests were two-tailed, a *p* < 0.05 was considered significant.

The area under the curve (AUC) after drug injections until the completion of the study were calculated using the trapezoidal method [40].

In the first study, we addressed whether the administration of oximes alone had any effects on the ventilation at rest. Accordingly, the effects of these treatments in rats were compared to the effects of the solvent (sodium isotonic solution) administered to animals using a two-way analysis of variance for repeated measurements. For each parameter, when a significant time*treatment interaction was found, this analysis was followed by multiple pair-wise comparisons.

In the second study, the aim was to study the antidotal activity of the oximes toward diethyl-paraoxon-induced respiratory toxicity. Accordingly, the oxime groups were compared to the diethyl-paraoxon group while the solvent group allowed assessing the magnitude of the reversal of paraoxon-induced respiratory toxicity. Statistical analyses were made using a two-way analysis of variance for repeated measurements. For each parameter, when a significant time*treatment interaction was found, this analysis was followed by multiple pair-wise comparisons.

## Figures and Tables

**Figure 1 molecules-25-03056-f001:**
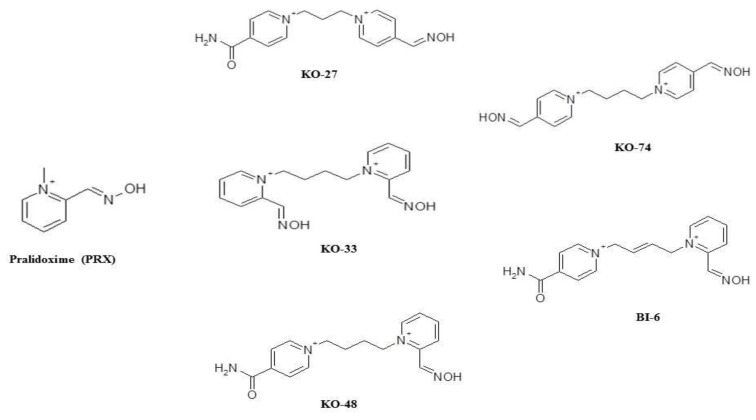
Chemical structure of pralidoxime (PRX) and new bisquaternary symmetric (KO-33, KO-74) or asymmetric pyridinium (KO-27, KO-48, BI-6) aldoximes.

**Figure 2 molecules-25-03056-f002:**
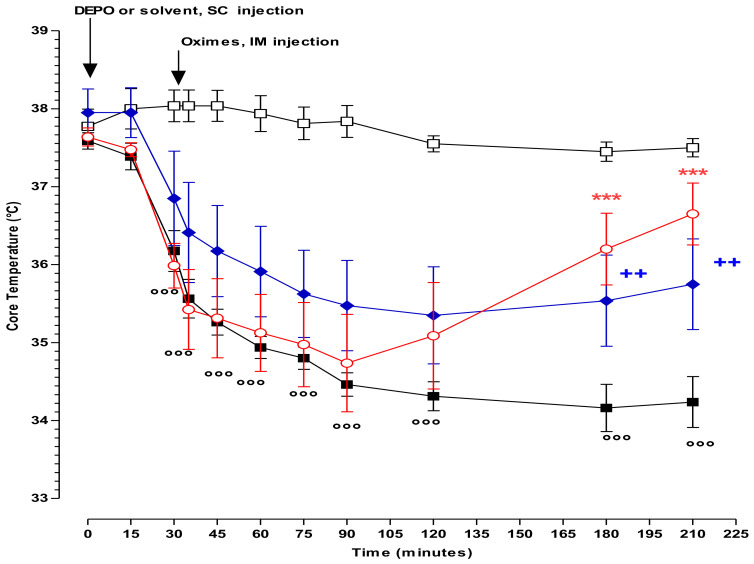
Effects of solvent (white square), diethyl-paraoxon (DEPO, black square), diethyl-paraoxon + pralidoxime (DEPO+PRX, blue diamond), and diethyl-paraoxon + BI-6 (DEPO+BI-6, open red circle) on core temperature. Arrows mark the injection time of solvent or DEPO and PRX or BI-6. Each group consisted of eight rats. Data are represented by mean ± S.E.M. at each time after injection. DEPO vs. Control: °°° *p* < 0.01; DEPO+BI-6 vs. PRX: ++ *p* < 0.001; DEPO+BI-6 vs. DEPO: *** *p* < 0.001.

**Figure 3 molecules-25-03056-f003:**
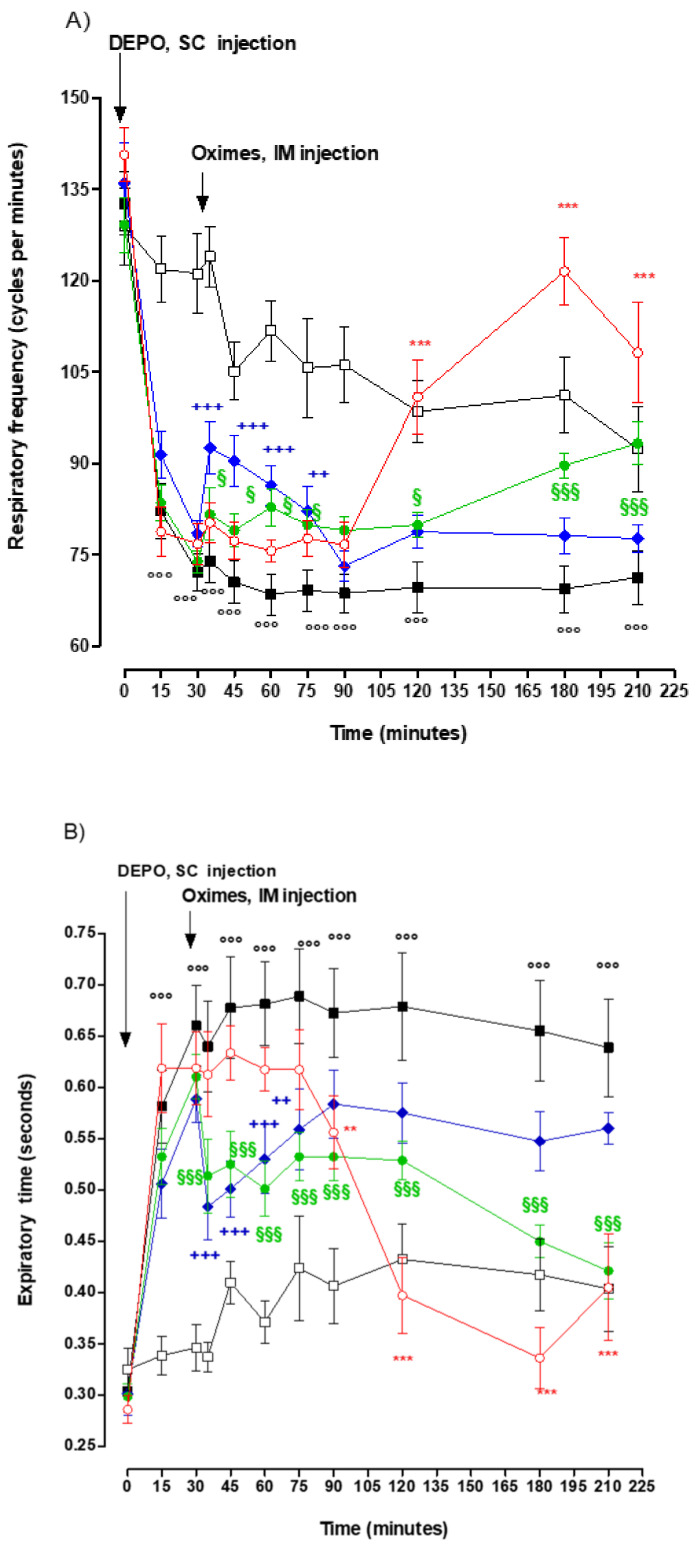
Time-course of frequency *(f*, Panel **A**), expiratory time (*T*_E_, Panel **B**), total time (*T*_TOT,_ Panel **C**), and tidal volume (*V*_T_, Panel **D**) in control group (open squares-black line), diethyl-paraoxon (DEPO) group (black squares-black line), diethyl-paraoxon + pralidoxime (DEPO+PRX group (blue diamond-blue line), DEPO + KO-27 group (green diamond-green line), and DEPO + BI-6 group (red open circle-red line). Each group contained eight rats. Data are represented by mean ± SEM. DEPO *vs* Control: °°°*p* < 0.001; DEPO + PRX vs. DEPO: ^++^
*p* < 0.01, ^+++^
*p* < 0.001; DEPO + KO-27 vs. DEPO: ^$$^
*p* < 0.01, ^$$$^
*p* < 0.001; DEPO+BI-6 vs. DEPO: ^**^
*p* < 0.01, ^***^
*p* < 0.001.

**Table 1 molecules-25-03056-t001:** Values of area under curve (AUC) from 0 to 210 min for core temperature. Each group consisted of five rats. Data are represented by mean ±S.E.M. at each time after injection.

	Control Group	Pralidoxime Group	KO-27 Group	KO-33 Group	KO-48 Group	KO-74 Group	BI-6 Group
**AUC** _**0-210min**_ **(Celsius.min)**	7919 ± 590	7855 ± 513	7976 ± 542	8102 ± 447	8023 ± 470	7915 ± 503	8050 ± 585

**Table 2 molecules-25-03056-t002:** Values of AUC from 0 to 210 min for different respiratory parameters. Each group consisted of five rats. Data are represented by mean **±** S.E.M. at each time after injection.

	Control Group	Pralidoxime Group	KO-27 Group	KO-33 Group	KO-48 Group	KO-74 Group	BI-6 Group
**AUC_0-210min_ frequency (cycles)**	17321 ± 4370	17374 ± 4638	17633 ± 4733	17132 ± 4857	17268 ± 4629	17720 ± 4748	17321 ± 4370
**AUC_0-210min_ total time (second.min)**	85 ± 23	78 ± 21	78 ± 21	79 ± 21	80 ± 21	76 ± 21	82 ± 22
**AUC_0-20mn_ expiratory time** **(second.min)**	61 ± 17	55 ± 15	54 ± 15	54 ± 15	54 ± 15	53 ± 14	58 ± 16
**AUC_0-210min_ inspiratory time** **(second.min)**	24 ± 6	23 ± 6	24 ± 7	25 ± 7	25 ± 7	23 ± 6	24 ± 6
**AUC_0-210min_ tidal volume (µL.min)**	217210 ± 58324	221008 ± 59493	216971 ± 58241	202218 ± 46189	217133 ± 58391	204421 ± 52174	221185 ± 59502
**AUC_0-210min_ minute ventilation (µL)**	23489827	25086912 ± 6699540	25629140 ± 6868978	21482414 ± 5751740	24726875 ± 6628236	22729783 ± 6063776	24721913 ± 6620647

**Table 3 molecules-25-03056-t003:** Values of AUC from 0 to 210 min for core temperature. Each group consisted of eight rats. Data are represented by mean ± S.E.M. at each time after injection. Diethyl-paraoxon (DEPO) vs. Control: *** *p* < 0.001. Diethyl-paraoxon (DEPO) + oximes vs. Control: +++ *p* < 0.001.

	Control Group	DEPO Group	DEPO+PRX Group	DEPO+KO-27 Group	DEPO+KO-33 Group	DEPO+KO-48 Group	DEPO+KO-74 Group	DEPO+BI-6 Group
**AUC_0-210min_** **(Celsius.min)**	7919 ± 640	7125 ± 513 ***	7245 ± 398 ^+++^	7076 ± 548 ^+++^	7035 ± 447 ^+++^	7243 ± 860 ^+++^	6915 ± 906 ^+++^	7250 ± 506 ^+++^

**Table 4 molecules-25-03056-t004:** Values of AUC from 0 to 210 min for respiratory parameters. Each group was constituted of eight rats. Data are represented by mean ±S.E.M. at each time after injection. Diethyl-paraoxon (DEPO) vs. Control: *** *p* < 0.001. Diethyl-paraoxon (DEPO) + oximes vs. Control: +++ *p* < 0.001.

	Control Group	DEPO Group	DEPO+PRX Group	DEPO+KO-33 Group	DEPO +KO-48 Group	DEPO+KO-74 Group
**AUC_0-210min_ frequency (cycles)**	22570 ± 6287	17315 ± 4280 ^***^	17839 ± 4982 ^+++^	16091 ± 4778 ^+++^	17068 ± 4352 ^+++^	17820 ± 4856 ^+++^
**AUC_0-210min_ total time (second.min)**	90 ± 34	65 ± 32 ^***^	80 ± 25 ^+++^	63 ± 18 ^+++^	68 ± 21 ^+++^	66 ± 21 ^+++^
**AUC_0-20mn_ expiratory time** **(second.min)**	68 ± 15	47 ± 9 ^***^	57 ± 6 ^+++^	43 ± 10 ^+++^	44 ± 9 ^+++^	43 ± 7 ^+++^
**AUC_0-210min_ inspiratory time** **(second.min)**	22 ± 8	18 ± 9	23 ± 8	20 ± 9	24 ± 10	23 ± 9
**AUC_0-210min_ tidal volume (µL.min)**	214231 ± 49325	20942 ± 48923	222008 ± 56723	204376 ± 41897	213422 ± 49391	23475 ± 51656
**AUC_0-210min_ minute ventilation (µL)**	23489827	25567785 ± 683214	25086912 ± 6699540	24765409 ± 5751740	24346784 ± 6628236	23678954 ± 6063776

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
