# Peer review of "Safety and Efficacy of New Oximes to Reverse Low Dose Diethyl-Paraoxon-Induced Ventilatory Effects in Rats"

_molecules, 2020, doi:10.3390/molecules25133056_

Round 1

Reviewer 1 Report

The aim of this study was to compare the safety and the 66 efficiency of selected new promising oximes including KO-27, KO-33, KO-48, KO-74 and BI-6 67 in comparison with those induced by pralidoxime in our rat model of non-lethal model of paraoxon toxicity. The safety study of oximes showed no adverse effects 24 on ventilation in rats. KO-33, KO-48, KO-74 oximes did not exhibit significant antidotal effect in 25 vivo. In contrast, KO-27 and BI-6 showed antidotal activity evidenced by normalization of 26 respiratory frequency and respiratory times. KO-27 became inefficient only during the last 30 min 27 of the study.

This study is interesting and well designed. Minor concerns exist:

-I suggest that abbreviations od DEPO, DEPO+PRX, etc. should be included in the legends of figures and tables since it is hard to move for other parts in order to find or memorize them

-Please correct the mistake in the following sentence: “In a previous study, we sho6wed the elimination half-life…”

-The calculation of equipotent doses of pralidoxime analogues is critical and how pralidoxime dose was selected. I kindly ask further explanations for the following: Determination of equipotent doses is quite difficult when comparing a mono-oxime to bioximes. Regarding pralidoxime, the referent dose in our studies was 50 mg/Kg of pralidoxime cation administered intra muscularly. To compare the efficiency of the same number of reactivation sites, we selected the dose 50 mg/Kg of each bi-oxime cation that have a molecular weight of about twice that of pralidoxime but providing two reactivation site per molecules.

-Was this study approved by an ethical committee for animal use in research?

Reviewer 2 Report

Organophosphate poisoning is still big threat especially in underdeveloped agriculture communities. Use of atropine together with oximes is still the best choice to decrease fatality. Authors' aim is to improved treatment of such poisonings with more efficient oximes. 

The results showed promising long term effects in two oximes, with no effects on basal values observed in this study. The safety study is inevitable, however as a starting point the results are promising.

Authors should pay more attention during proof reading of the manuscript, since there is a plenty of typos in the text (please see attached file). Different amount of carboxylesterase in the human vs rat should be taken in to consideration as well.

Overall the manuscript has a great potential to be interesting for readers.
